# Development of accurate and stable primary standard gas mixtures for global atmospheric acetonitrile monitoring: evaluating adsorption loss and long-term stability

Baigali Tsogt[1,2], Ji Hwan Kang[1], Seok-Young Oh[2], Sangil Lee[1,3]

[1]Gas Metrology Group, Korea Research Institute of Standards and Science, Daejeon, the Republic of Korea.
[2]Civil and Environmental Engineering Department, University of Ulsan, Ulsan, the Republic of Korea.
[3]Measurement Science, University of Science and Technology, Daejeon, the Republic of Korea.

*Correspondence to*: Sangil Lee (slee@kriss.re.kr)

**Abstract.** Acetonitrile plays an important role in atmospheric processes and serves as a key tracer of biomass burning, the major emission source of primary carbonaceous particles and trace gases in the global atmosphere. Recognizing its significance, the World Meteorological Organization (WMO) Global Atmosphere Watch (GAW) has designated acetonitrile as one of the target volatile organic compounds for global atmospheric monitoring, aiming for data quality better than 20 % accuracy and 15 % precision. Meeting the objectives of the WMO GAW program requires accurate and stable calibration standards with expanded uncertainties of less than 5 %. In this study, we explored the feasibility of gravimetry for developing primary standard gas mixtures (PSMs) in three different types of aluminium cylinders, each with distinct internal surface treatments, at nmol mol$^{-1}$ and µmol mol$^{-1}$ levels with a relative expanded uncertainty of less than 5 %, having nitrogen as matrix gas. We found that all three types of cylinders were inadequate without further passivation for developing acetonitrile PSMs below 10 nmol mol$^{-1}$ due to significant adsorption losses (6 % − 49 %) onto the cylinder's inner surface. To overcome this challenge, we prepared acetonitrile gas mixtures at 100 nmol mol$^{-1}$ using a modified gravimetric method and at 10 µmol mol$^{-1}$ using a conventional gravimetric method and then evaluated their long-term stability. Results showed that the effect of the adsorption loss at 100 nmol mol$^{-1}$ and 10 µmol mol$^{-1}$ was negated and negligible, respectively. Stability results show that we can disseminate acetonitrile calibration standards at both 100 nmol mol$^{-1}$ and 10 µmol mol$^{-1}$ with a relative expanded uncertainty of 3 % and 1 %, respectively (with an expiration period of 3 years), meeting the target uncertainty of the WMO GAW program. Further research is still needed to develop accurate and stable acetonitrile calibration standards below 10 nmol mol$^{-1}$ that are closer to atmospheric levels.

## 1 Introduction

Biomass burning is the largest source of primary fine carbonaceous particles and the second largest source of trace gases in the global troposphere (Akagi et al., 2011). Biomass burning also emits a significant amount of short-lived global warming substances such as volatile organic compounds (VOCs) and NO$_x$ that significantly contribute to ozone formation through

photochemical reactions (van der Werf et al., 2017; Chandra et al., 2020). These photochemical oxidants and fine particles can cause severe regional air pollution and contribute to climate change (Ramana et al., 2010). Acetonitrile ($CH_3CN$) predominantly released from biomass burning is relatively unreactive, but it plays an important role in atmospheric ion formation through its chemical reaction with hydroxyl (OH) radicals (Brasseur et al., 1983; Yuan et al., 2010). Acetonitrile with its relatively long atmospheric lifetime ranging from 3 months to 11 months has been used as a useful tracer of both controlled and uncontrolled biomass burning in many research studies (Hamm and Warneck, 1990; De Gouw et al., 2003; Singh et al., 2003; Murphy et al., 2010). The global budget of acetonitrile is incomplete and poorly constrained due to the limited availability of in-situ measurement data on the atmospheric distribution, with even less data in remote marine atmospheres (Harrison and Bernath, 2013). Monitoring its long-term trend is a useful way for analysing changes in both natural and anthropogenic acetonitrile emissions in the global atmosphere. Acetonitrile has been measured at various locations and reported at pmol $mol^{-1}$ levels (Sanhueza et al., 2001; Warneke and De Gouw, 2001; Sanhueza et al., 2004) but the measurement uncertainties are too large to allow comparison between different studies and to gain new insights about its roles in atmospheric chemical and physical processes. The World Meteorological Organization (WMO) Global Atmosphere Watch (GAW) programme has selected acetonitrile as one of the target VOCs that should be monitored in the atmosphere and set the objective of the data quality as better than 20 % accuracy and 15 % precision (WMO, 2006). To achieve this objective, accurate and stable acetonitrile calibration standards are required with a relative expanded uncertainty of less than 5 % (Schultz et al., 2015). Several types of calibration standards have been used to measure acetonitrile including a diluted gas standard from a permeation sources with measurement accuracy of 25 %, (Singh et al., 2003; Lange, 2002) which requires a permeation device to keep a very stable temperature (Susaya et al., 2011), a standard gas mixture in a cylinder with relative expanded uncertainty of 15 % (Sanhueza et al., 2004), and liquid solution standards with relative standard deviations of 3 % (Zamecnik and Tam, 1987). However, the relatively large uncertainties of these standards do not meet the WMO GAW requirements. Thus, an International System of Units (SI) traceable accurate and stable acetonitrile calibration standard is essential for monitoring its concentrations and temporal and spatial variabilities and thus understanding its roles in atmospheric processes and climate change. There are several methods such as static gravimetry (ISO, 2015), dynamic gravimetry (Brewer et al., 2011) and dynamic dilutions (Kim et al., 2016) for producing primary standard gas mixtures (PSMs). This study investigates the feasibility of gravimetry for developing acetonitrile PSMs with nitrogen matrix in three different types (i.e., different internal surface treatments) of aluminium cylinders at nmol $mol^{-1}$ and µmol $mol^{-1}$ levels with a relative expanded uncertainty of less than 5 % (a confidence interval of approximately 95 %, $k = 2$).

## 2 Materials and methods

### 2.1 Materials and preparation of gas mixtures

This study involves preparing gas mixtures using the conventional gravimetric method (ISO, 2015) and the modified gravimetric method (Brewer et al., 2019) to examine the stability of acetonitrile (ACN) in aluminium cylinders. Hexane known

to be stable in aluminium cylinders (Rhoderick et al., 2019) was introduced with acetonitrile to monitor the stability of acetonitrile in cylinders. In the conventional gravimetric method, an aliquot of a parent gas mixture was transferred to another new cylinder and then diluted with pure nitrogen to prepare gas mixtures at lower amount fractions (Fig. 1a,b). For the modified

gravimetric method, after decanting some amount of the aliquot, the remaining aliquot in a parent gas mixture is diluted with pure nitrogen which negates the effect of adsorption loss on the internal surface of cylinders (Brewer et al., 2019) (Fig. 1c). At each dilution step, at least two pairs of cylinders of mixtures were prepared, the exact number of cylinders is shown in the schematic diagram in Fig. 1.

Both acetonitrile and hexane liquid reagents (Sigma-Aldrich, USA) were analysed to assess their purity using a gas

chromatograph with a flame ionization detector (GC-FID, 7890N Agilent Technologies, USA) for VOC impurities and a Karl Fischer coulometer (831 KF Metrohm, Switzerland) for water impurities. The purity of the acetonitrile reagent was estimated as 99.9458 cmol mol$^{-1}$ with an expanded uncertainty of 0.0023 cmol mol$^{-1}$ ($k$ = 2). The purity of the hexane reagent was calculated as 99.2672 cmol mol$^{-1}$ with an expanded uncertainty of 0.0157 cmol mol$^{-1}$. No quantifiable acetonitrile was found in the hexane reagent liquid. High-purity nitrogen gases (Deokyang Co. Ltd, Korea) were used as a diluent and were purified

through a purifier (Micro Torr SAES Pure Gas Inc., USA) to remove VOCs to less than 1 nmol mol$^{-1}$. The acetonitrile content in high-purity nitrogen gases was determined to be half of its detection limit (0.31 nmol mol$^{-1}$) with an expanded uncertainty of 0.18 nmol mol$^{-1}$.

The acetonitrile and hexane liquid reagents were transferred into the cylinders using gastight syringes with a sample-lock termination fitted with a bevelled-tip stainless steel needle (Hamilton Company, USA). Each syringe was locked by a twist

valve on termination and its needle tip was capped with a septum to minimize the possible evaporation loss of liquid reagents during preparation. To determine the masses of liquid reagents transferred into cylinders, the syringes were weighed on an analytical balance (AT201 Mettler Toledo, Switzerland) with a capacity of 205 g and a resolution of 0.01 mg. To calculate the masses of nitrogen gases introduced into cylinders, the cylinders were weighed before and after being filled with gas using a KRISS automatic weighing system equipped with a top-pan balance (XP26003L Mettler Toledo, Switzerland) with a capacity

of 26 kg and a resolution of 1 mg.

Gravimetric preparation was used as the primary method to establish SI traceability through direct linkage to the mole (mol). The amount of substance in sample X, $n$, was determined following common and practical realizations of the mole definition and its derived units as shown in the following Eq. (1) (Güttler et al., 2019).

$$n = \frac{N}{N_A} = \frac{w(X)m}{A_r(X)N_A m_A} = \frac{w(X)m}{A_r(X)M_u} \tag{1}$$

Where

| | | |
|---|---|---|
| $N$ | the number of elementary entities of the substance X in the sample, | |
| $N_A$ | Avogadro constant | (mol$^{-1}$), |
| $w(X)$ | the mass fraction of X in the sample | (g g$^{-1}$), |
| $m$ | the mass of the $N$ elementary entities | (g), |

$A_r(X)$      the relative atomic or molecular mass of X (depending on whether X is an element or a compound respectively),

         $M_u$      the molar mass constant                         (g mol$^{-1}$).

Gas mixtures were prepared in three different types of 10 L aluminium cylinders (Luxfer, United Kingdom) with different internal surface treatments such as untreated cylinders with nickel chrome plated valves (Hamai, Japan) (hereafter referred to as Untreated), Experis-treated cylinders (Air Products, Belgium) with stainless steel valves (Rotarex, Luxembourg) (hereafter

referred to as Experis) and Performax-treated cylinders (EffecTech, United Kingdom) with stainless steel valves (Rotarex, Luxembourg) (hereafter referred to as Performax). All cylinders were new and vacuumed to $10^{-3}$ Pa ($10^{-5}$ mbar) while being heated to about 70 °C with temperature-controlled heating bands to ensure the thorough removal of any potential impurities prior to gravimetry preparation. Each dilution gas was introduced into cylinders using a KRISS gas filling system that consisted of a Sulfinert®-treated stainless steel manifold gate valves, a vacuum pump and pressure gauges (Kim et al., 2018).

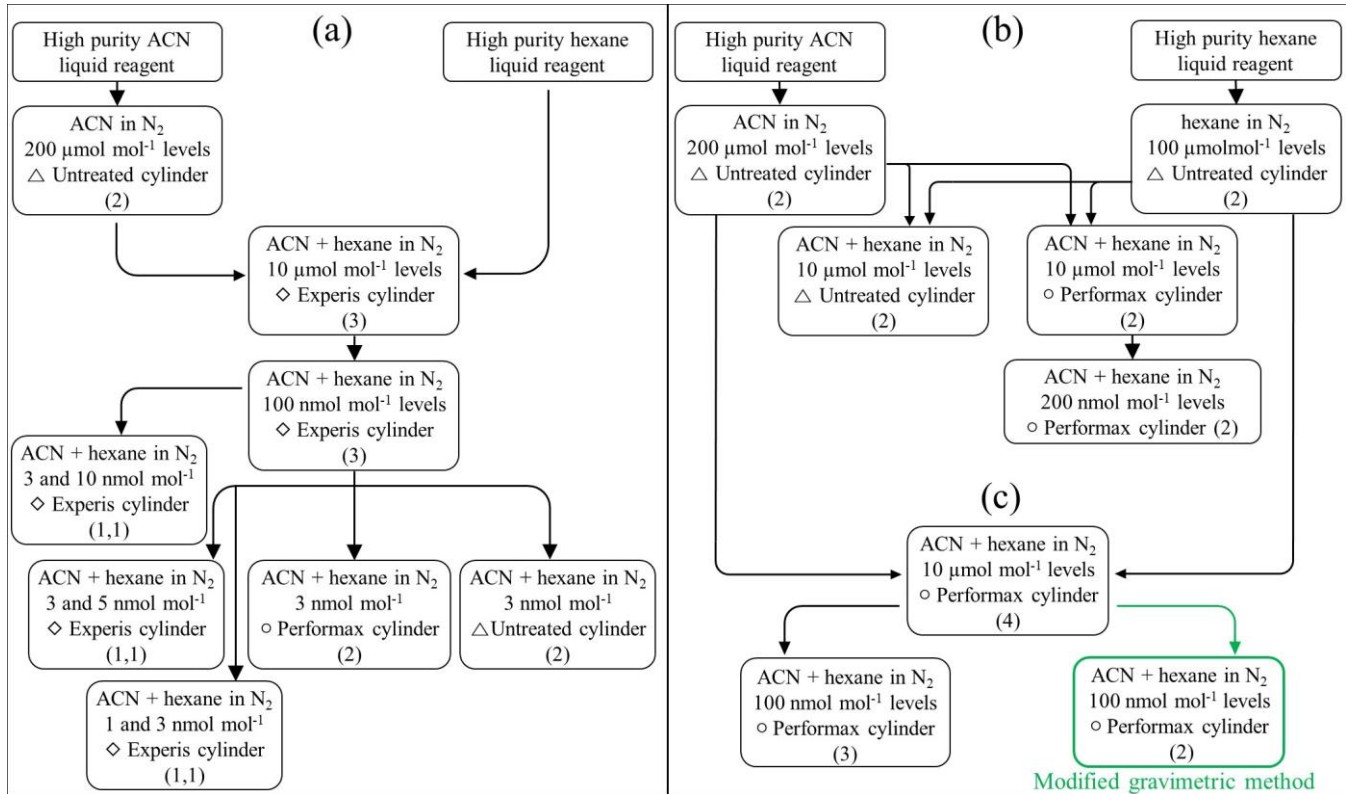

**Figure 1. Schematic diagram of the preparation of the acetonitrile gas mixtures with the target amount fractions in specified cylinders (a) by conventional gravimetric method, (b) by conventional gravimetric method for adsorption loss evaluation and (c) by modified gravimetric method. The number in parentheses indicates the number of cylinders prepared at each level.**

**2.2 Analysis of gas mixtures**

The GC-FID analytical instruments utilized in this study have been developed for analysing dimethyl sulphide at nmol mol$^{-1}$ levels and previously validated by Kim et al. (2016; 2018). They have been successfully employed for analysing volatile

organic compounds at nmol mol$^{-1}$ levels in multiple international comparisons (Lee et al., 2020; Lee et al., 2021; Cecelski et al., 2022; Lee et al., 2022) to demonstrate the measurement capabilities. The GC-FID (Agilent 7890, USA) equipped with a 2 µL sample loop was used for analysing samples in the range of 10 to 200 µmol mol$^{-1}$. For nmol mol$^{-1}$ range measurements, a GC-FID (Agilent 6890, USA) coupled with a cryogenic pre-concentrator was utilized. In the pre-concentration system, approximately 0.9 L for 1–10 nmol mol$^{-1}$ and 0.5 L for 100 nmol mol$^{-1}$ of sample gas, respectively, was trapped in a Sulfinert®-treated sample loop filled with glass beads under cryogenic conditions using liquid nitrogen. The more detailed configuration of the sample inlet to the cryogenic pre-concentrator has been previously described by Kim et al. (2018).

Both systems were equipped with DB-1 capillary column (60 m × 0.32 mm, 1 µm film thickness, Agilent, USA) using helium as the carrier gas. The GC oven temperature program consisted of an initial hold at 80 °C for 3 min followed by a ramp of 20 °C min$^{-1}$ to 150 °C with a final hold of 3 min. The FID temperature was maintained at 250 °C. Prior to each set of samples blank measurements using high-purity nitrogen (99.9999 cmol mol$^{-1}$) were conducted. For each preparation level, all gas mixtures were analysed against each other to evaluate the consistency of the gravimetrically prepared gas mixtures, with one of the gas mixtures selected as the working reference. Eight consecutive measurements (i.e., injections) were performed for each sample and peak areas were integrated baseline-to-baseline using the GC software. The averaged peak area for each sample was calculated using at least the last three measurements.

## 3 Results and discussion

### 3.1 Evaluation of gas mixtures less than 10 nmol mol$^{-1}$ prepared using conventional gravimetric method

We assessed the consistency of gas mixtures at each dilution step by comparing normalized response factors calculated following Eq. (4).

The response factor ($RF$) is determined by Eq. (2):

$$RF = \frac{y}{x} \tag{2}$$

Where

$y$       the analyser response (i.e., GC peak area),

$x$       the gravimetric amount fraction (mol mol$^{-1}$),

with the standard uncertainty of the response factor ($u(RF)$) given by:

$$u(RF) = \sqrt{u^2(y_{\text{rep}}) + u^2(y_{\text{drift}}) + u^2(x)} \tag{3}$$

Where

$u(y_{\text{rep}})$  the standard uncertainty of the repeatability of the analyser response (i.e., GC peak area),

$u(y_{\text{drift}})$ the standard uncertainty of the drift of the analyser response,

$u(x)$ the standard uncertainty of the gravimetric amount fraction (mol mol$^{-1}$).

The normalized response factor is determined by:

$$\text{Normalized } RF = \frac{RF_{\text{sample}}}{RF_{\text{reference}}} \tag{4}$$

with the standard uncertainty of the normalized response factor ($u(\text{Normalized } RF)$) given by:

$$u(\text{Normalized } RF) = \sqrt{u^2(RF_{\text{sample}}) + u^2(RF_{\text{reference}})} \tag{5}$$

Here $RF_{\text{sample}}$ and $RF_{\text{reference}}$ represent the response factor of sample and working reference, respectively, with their standard uncertainties $u(RF_{\text{sample}})$ and $u(RF_{\text{reference}})$, respectively. Blank measurements using high-purity nitrogen showed no detectable peaks, thus, blank correction was not required for the analytical results. The uncertainties of response factors were estimated by combining uncertainties from GC analysis and gravimetric preparation. If gas mixtures are consistent, the

150 normalized response factors should be consistent within their associated uncertainties (i.e., the normalized response factors should not be different from one within their associated uncertainties) considering a wide range of linearity of FID detectors. The normalized $RF$s of hexane in all cylinders agreed within the uncertainties (less than 0.5 %) regardless of cylinder types and amount fractions. However, the normalized $RF$s of acetonitrile showed inconsistency between cylinders with different treatments and even within cylinders with the same treatment (Fig. 2). For example, in both Experis and Untreated cylinders,

the normalized $RF$s of acetonitrile differed by about 10 % at the same amount fraction (3 nmol mol$^{-1}$) while those of hexane remained consistent. These results suggest that the inconsistency of acetonitrile resulted from cylinder characteristics (e.g. adsorption loss) related to acetonitrile, rather than from the gravimetry preparation itself, as hexane in the same cylinders showed good agreement. The normalized $RF$s for acetonitrile showed good agreement only across the Performax cylinders, despite the potential for some loss. Based on these findings, further tests and evaluations were focused on the Performax

cylinders.

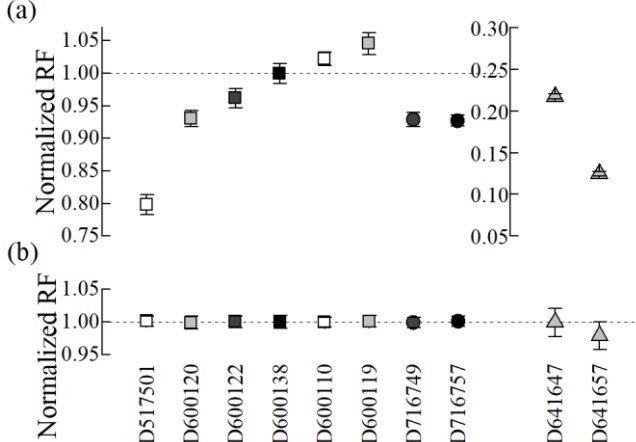

**Figure 2. Verification results of (a) acetonitrile and (b) hexane gas mixtures at 1 nmol mol$^{-1}$ (white), 3 nmol mol$^{-1}$ (grey), 5 nmol mol$^{-1}$ (dark grey), and 10 nmol mol$^{-1}$ (black) in the Experis (rectangle), Performax (circle), and Untreated (triangle) cylinders. Note that the y-axis and x-axis represents the normalized response factor (RF) and cylinder numbers, respectively, and the error bars show expanded uncertainties ($k = 2$).**

### 3.2 Evaluation of adsorption loss on the internal surface of cylinders

To estimate the adsorption loss of acetonitrile on the internal surface of the cylinder, we employed the cylinder-to-cylinder division method (Lee et al., 2017) for 3 nmol mol$^{-1}$ gas mixtures prepared in all three types of cylinders. The division method involved transferring a gas mixture from one cylinder to a new cylinder of the same type. Ideally, the peak area ratio of acetonitrile in the two cylinders should be equal to one within the analytical uncertainty, indicating little absorption loss. However, while the ratios of hexane remained equal to one for all cylinders, those of acetonitrile were notably lower, showing approximately 12 %, 22 % and 98 % decreases in peak areas in the Experis, Performax and Untreated cylinders, respectively (Fig. 3). This suggests about 6 %, 11 % and 49 % loss of acetonitrile during its gravimetric preparation, primarily attributed to substantial adsorption on the internal surface of the cylinders. This significant loss indicates that all three types of cylinders are unsuitable for preparing PSMs at amount fractions below 10 nmol mol$^{-1}$ without further proper passivation of the internal surfaces to minimize adsorption loss.

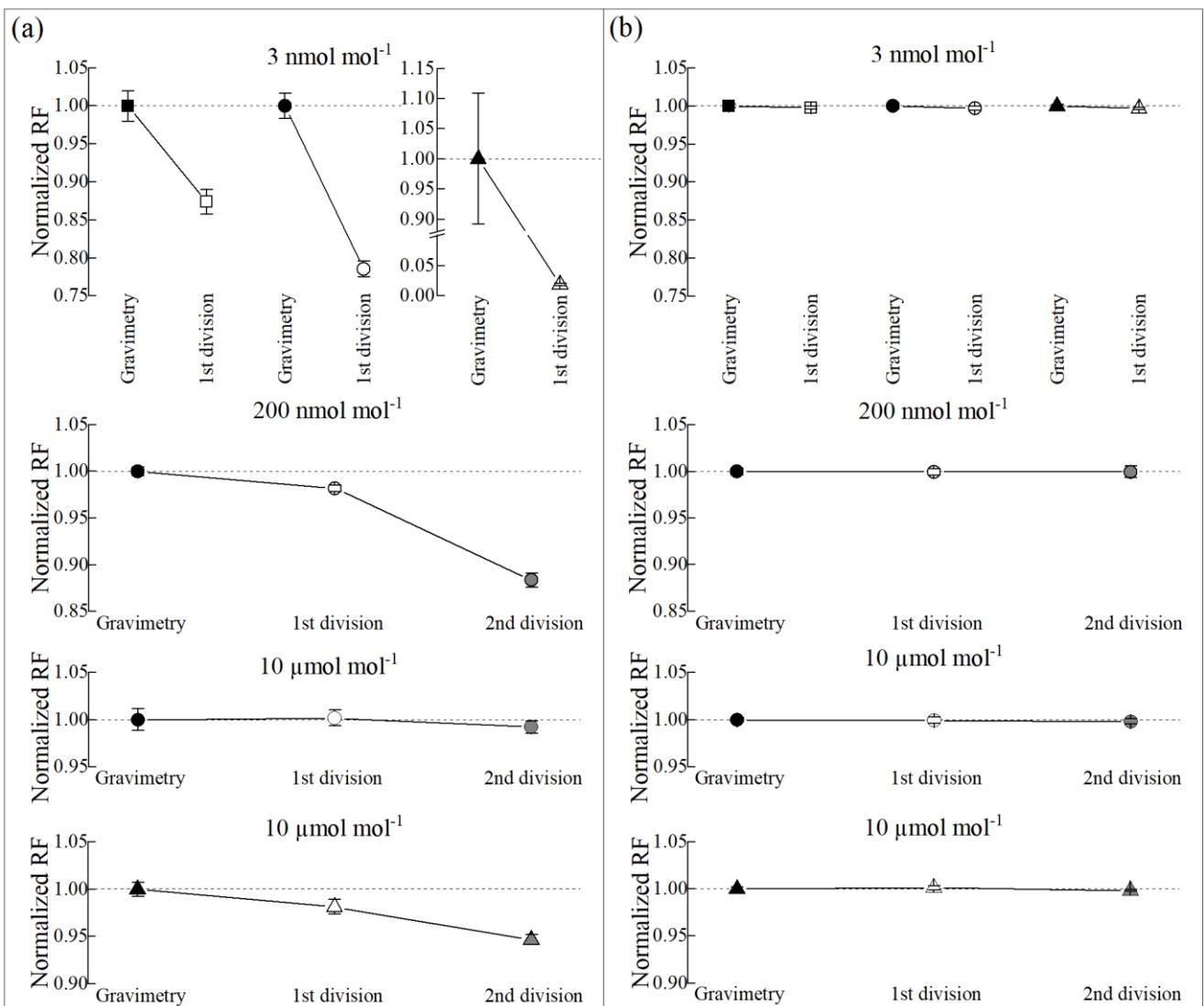

**Figure 3. Adsorption loss results of (a) acetonitrile and (b) hexane at various amount fractions in the Experis (rectangle), Performax (circle) and Untreated (triangle) cylinders. Note that the y-axis and x-axis represents the normalized response factor (RF) and cylinder numbers, respectively, and the error bars show expanded uncertainties ($k = 2$).**

Furthermore, we prepared additional gas mixtures exclusively in the Performax cylinders at 200 nmol mol$^{-1}$ about 70 times higher amount fraction than 3 nmol mol$^{-1}$, to evaluate whether the adsorption loss was negligible at such a higher amount fraction, given that acetonitrile at 3 nmol mol$^{-1}$ was consistent in only Performax cylinders despite its loss (Fig. 2). However, we still observed about 0.9 % (at the first division) and 5 % (at the second division) adsorption loss of acetonitrile at 200 nmol mol$^{-1}$. While relative adsorption losses decreased compared with 11 % at 3 nmol mol$^{-1}$, they were still larger than analytical uncertainties. To further investigate, we conducted additional tests using 10 μmol mol$^{-1}$ (more than 3000 times higher than 3 nmol mol$^{-1}$) gas mixtures. Thus far in the study, we developed 10 μmol mol$^{-1}$ gas mixtures in all three types of cylinders,

previously prepared Experis cylinders (Fig. 1a) and additionally prepared Performax and Untreated cylinders (Fig. 1b). The consistency test was conducted on all three types of cylinders alongside, and results showed that $RF$s of the Experis cylinders were about 3 % to 6 % lower than those of Performax and Untreated cylinders for acetonitrile, however hexane in all cylinders showed good agreement. As observed in the prior test results of gas mixtures less than 10 nmol mol$^{-1}$, acetonitrile $RF$s were not consistent across all three Experis cylinders at even 10 µmol mol$^{-1}$, indicating that the adsorption loss persists at higher levels and the amount of loss varies from cylinder to cylinder. Therefore, Experis cylinders were excluded from the adsorption loss evaluation, and adsorption loss tests were conducted on other two types of cylinders. The results showed little loss, which is less than its associated relative analytical uncertainties (about 0.6 % for acetonitrile and 0.2 % for hexane), concerning both acetonitrile and hexane in the Performax cylinders. In contrast, approximately 1 % to 2 % loss of acetonitrile was observed in the Untreated cylinders. The results indicated that, while the effect of the adsorption loss at 10 µmol mol$^{-1}$ was nullified for acetonitrile in the Performax cylinders, it persisted in the Untreated cylinders.

### 3.3 Evaluation of gas mixtures at 100 nmol mol$^{-1}$ prepared using modified gravimetric method

Adsorption loss was nullified only in Performax cylinders at 10 µmol mol$^{-1}$, while significant acetonitrile loss was observed in both Experis and Untreated cylinders at the same amount fraction. Therefore, acetonitrile gas mixtures at 100 nmol mol$^{-1}$ were prepared in Performax cylinders using a modified gravimetric method. For comparison, two additional gas mixtures were prepared in Performax cylinders using the conventional gravimetric method. The consistency of four gas mixtures (i.e., two for each method) was evaluated by comparing the normalized $RF$s. Hexane exhibited consistency across all four cylinders, with a relative expanded uncertainty of 0.5 % ($k = 2$) regardless of the preparation method. However, acetonitrile showed consistency only within each method (Fig. 4). Furthermore, the normalized $RF$s of the gas mixtures prepared using the modified method were approximately 5 % higher than those prepared using the conventional method. This discrepancy can be attributed to the fact that, unlike the conventional method, the modified method negated the adsorption loss on the internal surface of cylinders. Thus, these results suggest that the modified method may be preferable for preparing acetonitrile PSMs at 100 nmol mol$^{-1}$ rather than the conventional method.

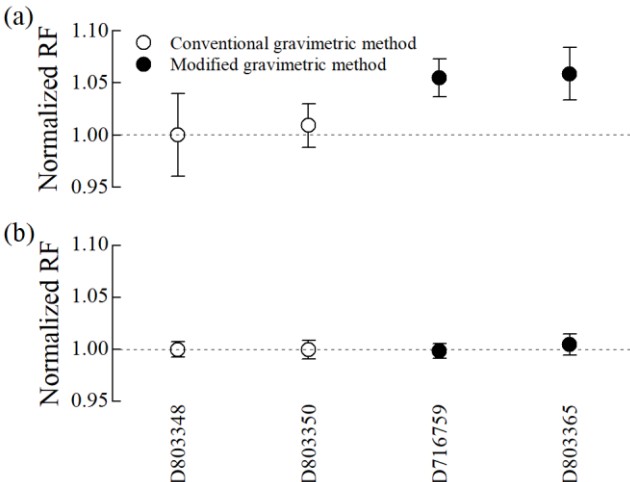

**Figure 4. Verification results of the 100 nmol mol⁻¹ (a) acetonitrile and (b) hexane in Performax cylinders. Note that the y-axis and x-axis represents the normalized response factor (RF) and cylinder numbers, respectively, and the error bars show expanded uncertainties ($k = 2$).**

### 3.4 Long-term stability of acetonitrile PSMs at 100 nmol mol⁻¹

When developing a PSM, ensuring its long-term stability is essential for disseminating calibration standards and monitoring both the long-term trend and the variability of acetonitrile in the atmosphere. The long-term stability of gas mixtures prepared using the modified method was assessed through two approaches: comparing response ratios (i.e., peak area ratios) of acetonitrile to hexane in each cylinder for 10 months (Fig. 5a) and comparing $RF$s of new cylinders to those of old cylinders over 3 years (Fig. 5b). For the 10-month stability assessment, the response ratios of acetonitrile to hexane in each of the two cylinders were in agreement within a relative expanded uncertainty of 2.2 % ($k = 2$), indicating stable acetonitrile levels for 10 months (Fig. 5a). In the case of the 3-year stability, two new gas mixtures prepared in 2022 were compared with an old gas mixture prepared in 2019. The normalized $RF$s of acetonitrile in both old and new cylinders showed inconsistency within a relative expanded uncertainty of 0.6 % ($k = 2$), with a difference of about 2.5 %. This difference exceeded the analytical uncertainties indicating a gradual decrease in acetonitrile levels at 100 nmol mol⁻¹ in Performax cylinders over 3 years. Although the 3-year stability of acetonitrile meets the target uncertainty (5 %) of the WMO GAW programme, further studies (e.g., improved passivation methods) are necessary to develop more accurate and stable acetonitrile PSMs at nmol mol⁻¹ levels.

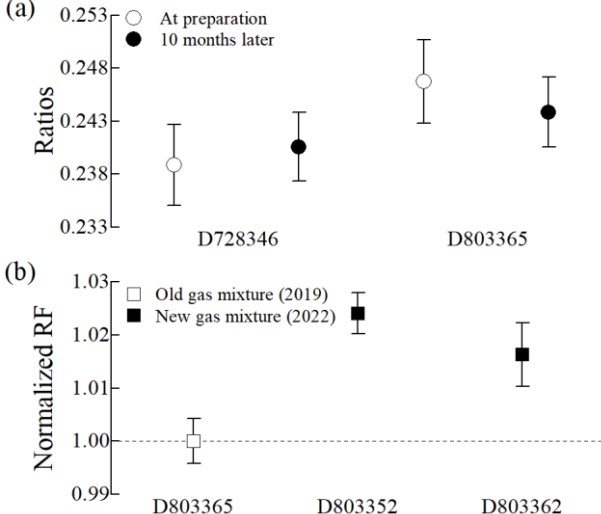

**Figure 5.** Long-term stability of the 100 nmol mol$^{-1}$ acetonitrile in the Performax cylinders (a) 10 month-stability (peak area ratios of hexane to acetonitrile), (b) 3 year-stability (the *RF*s of new gas mixtures are normalized to that of an old gas mixture). Note that the y-axis and x-axis represents the normalized response factor (RF) and cylinder numbers, respectively, and the error bars show expanded uncertainties ($k = 2$).

### 3.4 Long-term stability of acetonitrile PSMs at 100 nmol mol$^{-1}$

The results from the previous section indicated that the modified method effectively addressed the issue of adsorption loss on the inner surface of cylinders. However, despite this improvement, there was a gradual decrease in the amount fraction of acetonitrile at 100 nmol mol$^{-1}$ (Fig. 5b). To further assess the long-term stability, we evaluated the parent gas mixtures (10 µmol mol$^{-1}$) of 100 nmol mol$^{-1}$ acetonitrile gas mixtures in Performax cylinders (Fig. 1b) by comparing the *RF*s of new gas mixtures against an old one (D726463) to assess the three-year stability. The normalized *RF*s in all four gas mixtures (old and new ones) were consistent within their associated relative expanded uncertainties of 0.90 % ($k = 2$) (Fig. 6). These results indicated that 10 µmol mol$^{-1}$ acetonitrile in Performax cylinders remained stable for about three years.

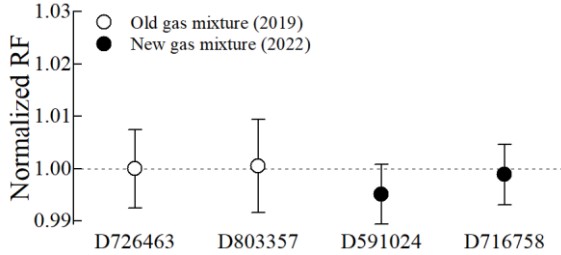

**Figure 6.** Long-term stability of the 10 µmol mol$^{-1}$ acetonitrile in the Performax cylinders. Note that the y-axis and x-axis represents the normalized response factor (RF) and cylinder numbers, respectively, and the error bars show expanded uncertainties ($k = 2$).

## 4 Conclusions and outlook

To support global atmospheric monitoring and enhance our understanding of the atmospheric processes and the roles of acetonitrile in air quality and climate changes, we evaluated the feasibility of three different types of cylinders for preparing gravimetric gas mixtures and then developed acetonitrile primary standard gas mixtures (PSMs). This included evaluating adsorption loss and long-term stability of acetonitrile in aluminium cylinders with three different surface treatments at both nmol mol$^{-1}$ and µmol mol$^{-1}$ levels. Our study revealed that all three types of cylinders tested in this study were unsuitable for

developing acetonitrile PSMs below 10 nmol mol$^{-1}$ due to significant loss (from 6 % to 48 %) on the inner surface of cylinders during the preparations unless additional surface treatments were developed and applied. The modified gravimetric method effectively mitigated adsorption loss at 100 nmol mol$^{-1}$, resulting in good consistency among gas mixtures prepared in Performax cylinders, which remained consistent after 10 months. A subsequent stability study based on the 10-months stability showed a gradual decrease of approximately 2.5 % over three years, with a decrease rate of approximately 0.9 % per year,

indicating that acetonitrile is predicted to be stable for 3 years with a relative expanded uncertainty of 3 %. For 10 µmol mol$^{-1}$ acetonitrile gas mixtures in Performax cylinders, we observed little adsorption loss, good consistency and 3-years long-term stability, making them suitable as PSMs. Based on these findings, acetonitrile calibration standards at 100 nmol mol$^{-1}$ with a relative expanded uncertainty of 3 % (meeting the target uncertainty of the WMO GAW programme) and at 10 µmol mol$^{-1}$ with a relative expanded uncertainty of 1 % can be disseminated with an expiration period of at least 3 years for global

atmospheric measurements. However, it is important to note that the amount fractions of acetonitrile calibration standards that can be disseminated is much higher than atmospheric levels, particularly in remote background areas. Therefore, further studies are needed to explore alternative methods, such as dynamic dilution and dynamic gravimetric methods, to generate accurate and stable acetonitrile gas mixtures at levels below 10 nmol mol$^{-1}$ more representative of atmospheric concentrations. In addition, future investigations can explore the use of electropolished stainless steel cylinders and Sulfinert® (i.e., SilcoNert®

2000) treatment as a passivation option. The silicon-based barrier in Sulfinert® treatment is chemically inert to most organic compounds (Barone et al., 2011; Vaittinen et al., 2013) and its non-polar surface could reduce the interaction with polar groups (Morriss and Isbister, 1986) such as acetonitrile which is moderately polar molecule (Zarzycki et al., 2010). Although the internal surface area of the valves was much less than that of the cylinders, it could be worth evaluating the impacts of different valve materials and coating methods for developing acetonitrile gas mixtures at nmol mol$^{-1}$ levels.

*Author contribution.* SL designed the experiments, and both JHK and BT carried them out. BT and SL prepared the manuscript with contributions from all co-authors.

*Competing interests.* The contact author has declared that none of the authors has any competing interests.

*Acknowledgements.* This research was supported by the Establishment of Measurement Standards for Greenhouse Gases and

Carbon Neutrality in Response to Climate Crisis funded by Korea Research Institute of Standards and Science (KRISS-2024-GP2024-0006-03).

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
