# Peer review of "Development of accurate and stable primary standard gas mixtures for global atmospheric acetonitrile monitoring: evaluating adsorption loss and long-term stability"

_Atmospheric Measurement Techniques, 2024_

## Referee Comment (RC1)

Comments by Maitane Iturrate-Garcia

The manuscript entitled "Development of accurate and stable primary standard gas mixtures for global atmospheric acetonitrile monitoring: evaluating adsorption loss and long-term stability" by Tsogt et al. describes two approaches that were followed to develop primary standard gas mixtures (PSM) of acetonitrile at amount of substance fractions relevant for atmospheric monitoring. The lack of stable primary standards – mainly due to surface adsorption losses – is a common problem for many gas compounds (e.g., VOCs such as methanol and acetaldehyde among others), which compromises the comparability of data collected by monitoring networks. Therefore, this work will be an important metrological contribution to the monitoring community.

**General comments**

The methods followed to prepare the primary standard gas mixtures are clearly described and the schematic provided increases the clarity of the preparation process. However, the manuscript will benefit from adding some relevant information and details regarding the standards *per se* (e.g., number of standards prepared in each type of cylinder) and the analytical methodology (e.g., GC/FID method followed: oven temperature ramps), as well as a broader discussion of the results found.

1. Are the produced primary standard gas mixtures traceable? If yes, to what? (e.g., traceability to the international system of units (SI), to KRISS). How was traceability achieved? This information is not detailed in the text, although in lines 47–49 the importance of SI-traceability is highlighted.

2. The number of cylinders used in this work should be clearly indicated. In the manuscript it is mentioned that three different types of cylinders were used but for the reader it is not clear if only one cylinder or several cylinders of each type were evaluated. It would be also helpful to provide information about the history of the cylinders: were they new cylinders or were they used previously for other compounds? If the latter, which compounds were present in the cylinders? Did the authors apply any conditioning procedure to the cylinders before the filling?

3. In the section materials and methods, it would be easier for the reader if subsections 2.1 'Materials' (lines 55–77) and 2.2.1 'Preparation of gas mixtures' (lines 78–90) are merged to one. As it is now, it is difficult to understand certain parts of the text, for example, why liquid reagents were selected and evaluated (lines 56-60). Presenting first 2.2.1 and then 2.1 will make the process clearer.

4. A more detail description of the gas mixtures analysis will contribute to the understanding of the results and to their reproducibility and reusability. Was the blank of the system assessed? What was the measurement sequence used? Was the GC/FID calibrated? How? How many replicates were measured of each sample? Were blanks measured within each measurement sequence? What was the GC/FID method used (e.g., oven temperature method: ramps, holding times…)?

5. Information about the way the uncertainty of the standards was estimated (e.g., uncertainty sources, model equations) should be included.

6. To improve the understanding of this work, the authors should consider to add the reasons why some of the tests were not performed on the three types of cylinders (e.g., line 139, line 145).

7. For comparison of the results among cylinders, it will be easier if the plots are presented with the same y-axis scale (e.g., Figure 2 (lines 116–117), Figure 1 (lines 130–131).

8. In lines 196–199, the authors stated that they can disseminate acetonitrile calibration standards with an expiration period of at least 3 years. As it is described, it seems that the

expiration period applies to both standards (100 nmol mol$^{-1}$ and 10 µmol mol$^{-1}$). However, in lines 192–193, it is indicated that the standard at 100 nmol mol$^{-1}$ in the Performax cylinder remained stable for 10 months. Some clarification and/or rephrasing might be needed.

9. The conclusions section would be more useful for the metrological and monitoring communities if instead of just summarising the findings of the work, some recommendations for further research to go forward with the gravimetry method for acetonitrile would be also added. For example, do the authors expect a different performance of the cylinders with the same treatment description but different valve material and/or coated valves? Do the authors have recommendations on potential passivation methods? Passivation is mentioned in lines 16, 128 and 168-169 but without giving details on the passivation method.

**Specific comments**

1. Line 13: does the value 5 % refer to standard uncertainties or to expanded uncertainties?

2. Lines 35–36: what is special with 'remote marine atmospheres' regarding the incomplete and poorly constrained global budget of acetonitrile?

3. Information about the gas matrix of the PSMs (Line 61) should appear earlier in the text (abstract, Line 51).

4. Line 76–77: it would be good to have more information on the models and manufacturers of the elements of the KRISS gas filling system (or a reference where the system is described).

5. Figure 1 (between lines 88–90): the abbreviation ACN, which is used for the first time in the text, should be introduced in the caption of Figure 1. Adding a frame to the schematic of each method would make the separation between (a) and (b) clearer.

6. Lines 92–97: Missing information on the manufacturer of DB-1 capillary column. How many replicates of each gas mixture was analysed?

7. Equation (1): what was used as reference?

8. Equations (1) and (2): were the peak areas corrected by the system blank?

9. Lines 117-118: the meaning of the different colours (white, grey, black) should be added to the caption of Figure 2, as well as the meaning of 'RF'. Does the second y-axis on the top panel indicate also the normalized RF?

10. A reference or description of the cylinder-to-cylinder division method 29 (lines 120–121) should be added.

11. Line 130: the figure numberings should be revised (currently, there are two Figures 1 and none Figure 3). What is the explanation for the large uncertainty of the black triangle point? For comparison purposes, it would be good to plot also the results of the internal standard (hexane). In this figure, results for cylinders at 200 nmol mol$^{-1}$ are shown; it would be good if these cylinders are included in the schematic of Figure 1. Were these cylinders prepared by the conventional or the modified gravimetric method? Why were only the Performax cylinders used for the gas mixtures at 200 nmol mol$^{-1}$? It would be good to add results of the untreated cylinders at 10 µmol mol$^{-1}$.

12. Line 139: what is meant by 'little loss'? Can some figure be given (e.g., < 0.1 %)?

13. What are the x-axis labels of Figures 4–6? It would be good if the authors indicate that they refer to the cylinder number. Adding a table with the cylinder number, the type of treatment and the amount fraction of the cylinder might make the text and figures clearer.

**Technical corrections**

Overall: missing space between numbers and '%'

Line 17: The hyphen used in the range (6%-49%) must be replaced by an en dash

Line 28: The character 'x' must be a subscript in $NO_x$

Line 30: The digit 3 must be a subscript in the acetonitrile formula ($CH_3CN$)

Line 52: 'nmol/mol' and 'μmol/mol' must be written as nmol $mol^{-1}$ and μmol $mol^{-1}$

Line 53: The symbol 'k' must be introduced (coverage factor $k$) and its font style changed to italic.

Line 63: 'nmol/mol' must be written as nmol $mol^{-1}$

Line 86: 'nmol/mol' must be written as nmol $mol^{-1}$

Line 94: Missing verb in the sentence 'For nmol mol-1 GC/FID with a cryogenic (liquid nitrogen) pre-concentrator used. '

Lines 120–121: Missing space between 'method' and '29'

Line 121: 'nmol/mol' must be written as nmol $mol^{-1}$

Line 143: the digit '-1' must be a superscript (nmol $mol^{-1}$)

Line 149: The font style of $RF$ should be changed to normal for consistency.

Line 167: the digit '-1' must be a superscript (nmol $mol^{-1}$)

Figures 1 and 2: 'nmol/mol' and 'μmol/mol' must be written as nmol $mol^{-1}$ and μmol $mol^{-1}$

---

## Author Response (AR1)

**RC1: Comments by Maitane Iturrate-Garcia**

Thank you for taking the time to review our research and provide such insightful and thorough feedback. Your comments are invaluable and will help strengthen our work. We particularly appreciate your insights and the careful attention you gave to our manuscript. We have incorporated your suggestions into the revised manuscript. Thank you again for your time and expertise.

General comments

1.    Are the produced primary standard gas mixtures traceable? If yes, to what? (e.g., traceability to the international system of units (SI), to KRISS). How was traceability achieved? This information is not detailed in the text, although in lines 47–49 the importance of SI-traceability is highlighted.

   **Authors' response**: The authors added more information regarding how to realize the definition of the mole, one of the seven SI base units in the page 3, lines 85–95, as shown in the following:

   "Gravimetric preparation was used as the primary method to establish SI traceability through direct linkage to the mole (mol). The amount of substance in sample X, $n$, was determined following common and practical realizations of the mole definition and its derived units as shown in the following Eq. (1) (Güttler et al., 2019).

   $$n = \frac{N}{N_A} = \frac{w(X)m}{A_r(X)N_A m_A} = \frac{w(X)m}{A_r(X)M_u} \tag{1}$$

   Where
   $N$     the number of elementary entities of the substance X in the sample,
   $N_A$    Avogadro constant                                      $(mol^{-1})$,
   $w(X)$  the mass fraction of X in the sample                     $(g\ g^{-1})$,
   $m$     the mass of the $N$ elementary entities                  $(g)$,
   $A_r(X)$ the relative atomic or molecular mass of X (depending on whether X is an element or a compound respectively),
   $M_u$    the molar mass constant                                 $(g\ mol^{-1})$."

2.    The number of cylinders used in this work should be clearly indicated. In the manuscript it is mentioned that three different types of cylinders were used but for the reader it is not clear if only one cylinder or several cylinders of each type were evaluated.
It would be also helpful to provide information about the history of the cylinders: were they new cylinders or were they used previously for other compounds? If the latter, which compounds were present in the cylinders? Did the authors apply any conditioning procedure to the cylinders before the filling?

   **Authors' response**: The authors added the number of cylinders in revised Figure 1 (page 4, lines 104–105), and added history about cylinders in the page 4, lines 99–102, as shown in the following.

   Original manuscript:

   "…and Performax-treated cylinders (EffecTech, United Kingdom) with stainless steel valves (Rotarex, Luxembourg) (hereafter referred to as Performax) …"

   Revised manuscript (page 4, lines 99–102):

   "…and Performax-treated cylinders (EffecTech, United Kingdom) with stainless steel valves (Rotarex, Luxembourg) (hereafter referred to as Performax). All cylinders were new and

vacuumed to $10^{-3}$ pa while being heated to about 70 °C with temperature-controlled heating bands to ensure the thorough removal of any potential impurities prior to gravimetry preparation."

Revised Figure 1:
[Figure]

[Figure]

**Figure 1. Schematic diagram of the preparation of the acetonitrile gas mixtures with the target amount fractions in specified cylinders (a) by conventional gravimetric method, (b) by conventional gravimetric method for adsorption loss evaluation and (c) by modified gravimetric method. The number in parentheses indicates the number of cylinders prepared in each level.**

**3.** In the section materials and methods, it would be easier for the reader if subsections 2.1 'Materials' (lines 55–77) and 2.2.1 'Preparation of gas mixtures' (lines 78–90) are merged to one. As it is now, it is difficult to understand certain parts of the text, for example, why liquid reagents were selected and evaluated (lines 56–60). Presenting first 2.2.1 and then 2.1 will make the process clearer.

**Authors' response**: To make the manuscript section clearer, the authors merged the part subsections 2.1 'Materials' and 2.2.1 'Preparation of gas mixtures' to '2.1 Materials and preparation of gas mixtures' in the page 2, lines 58–103, shown in the following:

Original manuscript:

"2.1 Materials

Both acetonitrile and hexane liquid reagents (Sigma-Aldrich, USA) were analysed to assess their purity using a gas chromatograph with a flame ionization detector (GC/FID; 7890N, Agilent Technologies, USA) for VOC impurities and a Karl Fischer coulometer (831 KF, Metrohm, Switzerland) for water impurities. The purity of the acetonitrile reagent was estimated…

2.2 Methods

2.2.1 Preparation of gas mixtures

A set of gas mixtures was prepared using both the conventional gravimetric method (ISO, 2015) and the modified gravimetric method (Brewer, 2011) (Fig. 1.). Hexane, known to be stable in aluminium cylinders (Rhoderick, 2013; Brewer, 2019), was introduced together with acetonitrile to monitor the stability of acetonitrile in cylinders…"

Revised manuscript (page 2, lines 58–103):

"2.1 Materials and preparation of gas mixtures

This study involves preparing gas mixtures using the conventional gravimetric method (ISO, 2015) and the modified gravimetric method (Brewer et al., 2019) to examine the stability of acetonitrile (ACN) in aluminium cylinders. Hexane known to be stable in aluminium cylinders (Rhoderick et al., 2019) was introduced with acetonitrile to monitor the stability of acetonitrile in cylinders. In the conventional gravimetric method…

Both acetonitrile and hexane liquid reagents (Sigma-Aldrich, USA) were analysed to assess their purity using a gas chromatograph with a flame ionization detector (GC-FID, 7890N Agilent Technologies, USA) for VOC impurities and a Karl Fischer coulometer (831 KF Metrohm, Switzerland) for water impurities. The purity of the acetonitrile reagent was estimated…"

**4.** A more detail description of the gas mixtures analysis will contribute to the understanding of the results and to their reproducibility and reusability. Was the blank of the system assessed? What was the measurement sequence used? Was the GC/FID calibrated? How? How many replicates were measured of each sample? Were blanks measured within each measurement sequence? What was the GC/FID method used (e.g., oven temperature method: ramps, holding times…)?

**Authors' response**: The authors revised the Subsection 'Analysis of gas mixtures' to provide more detailed description of the analysis in the page 4, lines 108–125. Moreover, more references regarding the analytical instruments were added in the manuscript (page 4, line 111–112). The GC-FIDs were not calibrated. They were used as a comparator to compare gas mixtures (prepared using gravimetry) against each other.

Original manuscript:

"2.2.2 Analysis of gas mixtures

All gas mixtures at each dilution step were analysed against each other (the first one is typically used as a reference) for verification at µmol mol-1 GC/FID with a DB-1 capillary column (60 m length and 0.32 mm diameter with a 1 µm thick film) and 2 µL loop utilized. For nmol mol$^{-1}$ GC/FID with a cryogenic (liquid nitrogen) pre-concentrator used. A sample gas at nmol mol-1 was cryogenically concentrated in a Sulfinert®-treated sample loop filled with glass beads. About a volume of 0.5 L sample was trapped in the loop and then injected into the GC/FID under the same conditions as µmol mol-1. The sample inlet to the cryogenic pre-concentrator was described in more detail in a previous study by Kim (2018)."

Revised manuscript (page 4, lines 108–125):

"2.2 Analysis of gas mixtures

The GC-FID analytical instruments utilized in this study have been developed for analysing dimethyl sulphide at nmol mol$^{-1}$ levels and previously validated by Kim et al. (2016; 2018). They have been successfully employed for analysing volatile organic compounds at nmol mol$^{-1}$ levels in multiple international comparisons (Lee et al., 2020; Lee et al., 2021; Cecelski et al., 2022; Lee et al., 2022) to demonstrate the measurement capabilities. The GC-FID (Agilent 7890, USA) equipped with a 2 µL sample loop was used for analysing samples in the range of 10 to 200 µmol mol$^{-1}$. For nmol mol$^{-1}$ range measurements, a GC-FID (Agilent 6890, USA) coupled with a cryogenic pre-concentrator was utilized. In the pre-concentration system, approximately 0.9 L for 1–10 nmol mol$^{-1}$ and 0.5 L for 100 nmol mol$^{-1}$ of sample gas, respectively, was trapped in a Sulfinert®-treated sample loop filled with glass beads under cryogenic conditions using liquid nitrogen. The more detailed configuration of the sample inlet to the cryogenic pre-concentrator has been previously described by Kim et al. (2018).

Both systems were equipped with DB-1 capillary column (60 m × 0.32 mm, 1 µm film thickness, Agilent, USA) using helium as the carrier gas. The GC oven temperature program consisted of an initial hold at 80 °C for 3 min followed by a ramp of 20 °C min$^{-1}$ to 150 °C with a final hold of 3 min. The FID temperature was maintained at 250 °C. Prior to each set of samples blank measurements using high-purity nitrogen (99.9999 cmol mol$^{-1}$) were conducted. For each preparation level, all gas mixtures were analysed against each other to evaluate the consistency of the gravimetrically prepared gas mixtures, with one of the gas mixtures was selected as the working reference. Eight consecutive measurements (i.e., injections) were performed for each sample and peak areas were integrated baseline-to-baseline using the GC software. The averaged peak area for each sample was calculated using at least the last three measurements."

**5.** Information about the way the uncertainty of the standards was estimated (e.g., uncertainty sources, model equations) should be included.

**Authors' response**: The authors revised the section regarding the uncertainty estimation as shown in the following:

Original manuscript:

"…assessed the consistency of gas mixtures at each dilution step by comparing normalized response factors (RFs) calculated following Eq. (1):

$$\text{Normalized } RF = \frac{RF_{sample}}{RF_{reference}} \tag{1}$$

and the response factor (RFi) determined by Eq. (2):

$$RF_i = \frac{GC\ peak\ area_i}{Gravimetric\ amount\ fraction_i} \tag{2}$$

Here, RFsample and RFreference represent the response factor of sample and reference, respectively. The uncertainties of response factors were estimated by combining uncertainties from GC analysis and gravimetric preparation…"

Revised manuscript (page 5, lines 128–143)

"We assessed the consistency of gas mixtures at each dilution step by comparing normalized response factors calculated following Eq. (4).

The response factor ($RF$) is determined by Eq. (2):

$$RF = \frac{y}{x} \tag{2}$$

Where

$y$      the analyser response (i.e., GC peak area),

$x$      the gravimetric amount fraction (mol mol$^{-1}$),

with the standard uncertainty of the response factor ($u(RF)$) given by:

$$u(RF) = \sqrt{u^2(y_{rep}) + u^2(y_{drift}) + u^2(x)} \tag{3}$$

Where

$u(y_{rep})$      the standard uncertainty of the repeatability of the analyser response (i.e., GC peak area),

$u(y_{drift})$      the standard uncertainty of the drift of the analyser response,

$u(x)$      the standard uncertainty of the gravimetric amount fraction (mol mol$^{-1}$).

The normalized response factor is determined by:

$$\text{Normalized } RF = \frac{RF_{sample}}{RF_{reference}} \tag{4}$$

with the standard uncertainty of the normalized response factor ($u(\text{Normalized } RF)$) given by:

$$u(\text{Normalized } RF) = \sqrt{u^2(RF_{sample}) + u^2(RF_{reference})} \tag{5}$$

Here $RF_{sample}$ and $RF_{reference}$ represent the response factor of sample and working reference, respectively, with their standard uncertainties $u(RF_{sample})$ and $u(RF_{reference})$, respectively…"

6. To improve the understanding of this work, the authors should consider to add the reasons why some of the tests were not performed on the three types of cylinders (e.g., line 139, line 145).

**Authors' response**: The authors added more detailed information responding to the reviewer's comment as shown in the following:

Original manuscript:

"…These results suggest that the inconsistency of acetonitrile resulted from cylinder characteristics (e.g., adsorption loss) concerning acetonitrile, rather than from the gravimetry preparation itself, as hexane in the same cylinders showed good agreement."

Revised manuscript (page 6, lines 155–158):

"…These results suggest that the inconsistency of acetonitrile resulted from cylinder characteristics (e.g. adsorption loss) related to acetonitrile, rather than from the gravimetry preparation itself, as hexane in the same cylinders showed good agreement. The normalized $RF$s for acetonitrile showed good agreement only across the Performax cylinders, despite the potential

for some loss. Based on these findings, further tests and evaluations were focused on the Performax cylinders."

Original manuscript:

"… To further investigate, we conducted addition tests using 10 µmol mol-1 (more than 3000 times higher than 3 nmol mol-1) gas mixtures in both Performax and Untreated cylinders. Results showed little loss for both acetonitrile and hexane in the Performax cylinders…"

Revised manuscript (page 8, lines 184–193):

"To further investigate, we conducted additional tests using 10 µmol mol$^{-1}$ (more than 3000 times higher than 3 nmol mol$^{-1}$) gas mixtures. Thus far in the study, we developed 10 µmol mol$^{-1}$ gas mixtures in all three types of cylinders, previously prepared Experis cylinders (Fig. 1a) and additionally prepared Performax and Untreated cylinders (Fig. 1b). The consistency test was conducted on all three types of cylinders alongside, and results showed that $RF$s of the Experis cylinders were about 3 % to 6 % lower than those of Performax and Untreated cylinders for acetonitrile, however hexane in all cylinders showed good agreement. As observed in the prior test results of gas mixtures less than 10 nmol mol$^{-1}$, acetonitrile $RF$s were not consistent across all three Experis cylinders at even 10 µmol mol$^{-1}$, indicating that the adsorption loss persists at higher levels and the amount of loss varies from cylinder to cylinder. Therefore, Experis cylinders were excluded from the adsorption loss evaluation, and adsorption loss tests were conducted on other two types of cylinders. The results showed little loss, which is less than its associated analytical uncertainties, for both acetonitrile and hexane in the Performax cylinders…"

**7.** For comparison of the results among cylinders, it will be easier if the plots are presented with the same y-axis scale (e.g., Figure 2 (lines 116–117), Figure 1 (lines 130–131).

**Authors' response**: As noted by the reviewer, the authors revised the scales in Figure 2 (lines 159–160), Figure 3 (lines 175–176), and Figure 4 (lines 209–210) to use the same y-axis scale. In contrast, in Figure 5 (lines 226–227) and Figure 6 (lines 239–240), the y-axis was scaled differently than in Figures 2–4 to clearly visualize the response factor variations of long-term stability and their uncertainty for each condition.

Revised Figure 2:

[Figure]

**Figure 2. Verification results of (a) acetonitrile and (b) hexane gas mixtures at 1 nmol mol$^{-1}$ (white), 3 nmol mol$^{-1}$ (grey), 5 nmol mol$^{-1}$ (dark grey), and 10 nmol mol$^{-1}$ (black) in the Experis (rectangle), Performax (circle), and Untreated (triangle) cylinders. Note that the y-axis and x-axis represents the normalized response factor (RF) and cylinder numbers, respectively, and the error bars show expanded uncertainties ($k = 2$).**

Revised Figure 3:

[Figure]

**Figure 3. Adsorption loss results of (a) acetonitrile and (b) hexane at various amount fractions in the Experis (rectangle), Performax (circle) and Untreated (triangle) cylinders. Note that the y-axis and x-axis represents the normalized response factor (RF) and cylinder numbers, respectively, and the error bars show expanded uncertainties ($k = 2$).**

Revised Figure 4:

[Figure]

**Figure 4. Verification results of the 100 nmol mol⁻¹ (a) acetonitrile and (b) hexane in Performax cylinders. Note that the y-axis and x-axis represents the normalized response factor (RF) and cylinder numbers, respectively, and the error bars show expanded uncertainties ($k = 2$).**

**8.** In lines 196–199, the authors stated that they can disseminate acetonitrile calibration standards with an expiration period of at least 3 years. As it is described, it seems that the expiration period applies to both standards (100 nmol mol⁻¹ and 10 µmol mol⁻¹). However, in lines 192–193, it is indicated that the standard at 100 nmol mol⁻¹ in the Performax cylinder remained stable for 10 months. Some clarification and/or rephrasing might be needed.

**Authors' response**: As noted by the reviewer, the authors revised the manuscript to improve clarity, as shown in the following:

Original manuscript:

"…effectively mitigated adsorption loss at 100 nmol mol⁻¹, resulting in good consistency among gas mixtures prepared in Performax cylinders, which remained stable for 10 months. Nevertheless, a subsequent stability study showed a gradual decrease (about 2.5% within less than three years after preparation) in acetonitrile levels with a decrease rate of approximately 0.9% per year… …Based on these findings, we can disseminate acetonitrile calibration standards at 100 nmol mol⁻¹ with a relative expanded uncertainty of 3% (meeting the target uncertainty of the WMO GAW programme), and at 10 µmol mol⁻¹ with a relative expanded uncertainty of 1% with an expiration period of at least 3 years for global atmospheric measurements."

Revised manuscript (page 12, lines 250–253):

"…effectively mitigated adsorption loss at 100 nmol mol⁻¹, resulting in good consistency among gas mixtures prepared in Performax cylinders, which remained consistent after 10 months. A subsequent stability study based on the 10-months stability showed a gradual decrease of approximately 2.5 % over three years, with a decrease rate of approximately 0.9 % per year, indicating that acetonitrile is predicted to be stable for 3 years with a relative expanded uncertainty of 3 %…"

…Based on these findings, acetonitrile calibration standards at 100 nmol mol⁻¹ with a relative expanded uncertainty of 3 % (meeting the target uncertainty of the WMO GAW programme) and at 10 µmol mol⁻¹ with a relative expanded uncertainty of 1 % can be disseminated with an expiration period of at least 3 years for global atmospheric measurements."

**9.** The conclusions section would be more useful for the metrological and monitoring communities if instead of just summarising the findings of the work, some recommendations for further research to go forward with the gravimetry method for acetonitrile would be also added. For example, do the authors expect a different performance of the cylinders with the same treatment description but different valve material and/or coated valves? Do the authors have recommendations on potential passivation methods? Passivation is mentioned in lines 16, 128 and 168–169 but without giving details on the passivation method.

> **Authors' response**: The authors revised the conclusions section, as per the reviewer's comment, by adding some recommendations on page 12, lines 261–267, as shown in the following:
>
> "… In addition, future investigations can explore the use of electropolished stainless steel cylinders and Sulfinert® (i.e., SilcoNert® 2000) treatment as a passivation option. The silicon-based barrier in Sulfinert® treatment is chemically inert to most organic compounds (Barone et al., 2011; Vaittinen et al., 2013) and its non-polar surface could reduce the interaction with polar groups (Morriss and Isbister, 1986) such as acetonitrile which is moderately polar molecule (Zarzycki et al., 2010). Although the internal surface area of the valves was much less than that of the cylinders, it could be worth evaluating the impacts of different valve materials and coating methods for developing acetonitrile gas mixtures at nmol mol$^{-1}$ levels."

**Specific comments**

**1.** Line 13: does the value 5 % refer to standard uncertainties or to expanded uncertainties?

**Authors' response**: As per the reviewer's comment, the value 5 % refers to expanded uncertainty and authors clarified the sentence as follows:

Original manuscript:

> "…calibration standards with uncertainties of less than 5%…"

Revised manuscript (abstract, lines 13–14):

> "…calibration standards with expanded uncertainties of less than 5 %..."

**2.** Lines 35–36: what is special with 'remote marine atmospheres' regarding the incomplete and poorly constrained global budget of acetonitrile?

**Authors' response**: The authors rephrased the paragraph to clarify the information in response to the reviewer's comment, as shown in the following:

Original manuscript:

> "…budget of acetonitrile is incomplete and poorly constrained due to the limited availability of in-situ measurement data on the atmospheric distribution, even in remote marine atmospheres (Harrison, 2013) …"

Revised manuscript (page 2, lines 36–37):

"…budget of acetonitrile is incomplete and poorly constrained due to the limited availability of in-situ measurement data on the atmospheric distribution, with even less data in remote marine atmospheres (Harrison and Bernath, 2013) …"

3. Information about the gas matrix of the PSMs (Line 61) should appear earlier in the text (abstract, Line 51).

**Authors' response:** As noted by the reviewer, the authors added the information about matrix gas in the abstract and the paragraph on page 2, line 54, as shown in the following:

Original manuscript:

"…three different types of aluminium cylinders, each with distinct internal surface treatments, at nmol mol$^{-1}$ and µmol mol$^{-1}$ levels with a relative expanded uncertainty of less than 5%…"

Revised manuscript (abstract, lines 15–16):

"…three different types of aluminium cylinders, each with distinct internal surface treatments, at nmol mol$^{-1}$ and µmol mol$^{-1}$ levels with a relative expanded uncertainty of less than 5 %, having nitrogen as matrix gas…"

Original manuscript:

"…This study investigates the feasibility of gravimetry for developing PSMs in three different types (i.e., different internal surface treatments) of aluminium cylinders…"

Revised manuscript (page 2, lines 53–55):

"…This study investigates the feasibility of gravimetry for developing acetonitrile PSMs with nitrogen matrix in three different types (i.e., different internal surface treatments) of aluminium cylinders…"

4. Line 76–77: it would be good to have more information on the models and manufacturers of the elements of the KRISS gas filling system (or a reference where the system is described).

**Authors' response**: The authors added the reference where the KRISS gas filling system is described, as suggested by the reviewer's comment.

Original manuscript:

"… All gases were introduced into cylinders using a KRISS gas filling system that consisted of a Sulfinert® -treated stainless steel manifold, gate valves, a vacuum pump, and pressure gauges."

Revised manuscript (page 4, lines 102–103):

"Each dilution gas was introduced into cylinders using a KRISS gas filling system that consisted of a Sulfinert®-treated stainless steel manifold gate valves, a vacuum pump and pressure gauges (Kim et al., 2018)."

The added reference as follows:

Kim M. E., Kang J. H., Kim Y. D., Lee D. S. and Lee S.: Development of accurate dimethyl sulphide primary standard gas mixtures at low nanomole per mole levels in high-pressure aluminium cylinders for ambient measurements. Metrologia, 55(2), 158, https://doi.org/10.1088/1681-7575/aaa583, 2018.

**5.**     Figure 1 (between lines 88–90): the abbreviation ACN, which is used for the first time in the text, should be introduced in the caption of Figure 1. Adding a frame to the schematic of each method would make the separation between (a) and (b) clearer.

**Authors' response:** As noted by the reviewer, the abbreviation ACN was introduced before the caption of Figure 1 (lines 104–105), on page 2, line 60. Additionally, the authors revised Figure 1 based on the comment to enhance clarity, with separation frame added between (a) and (b), as follows:

Revised paragraph (page 2, line 60):

"…gravimetric method (Brewer et al., 2019) to examine the stability of acetonitrile (ACN) in aluminium cylinders."

Original Figure 1:

**Figure 1. Schematic diagram of the preparation of the acetonitrile gas mixtures with the target amount fractions in specified cylinders, (a) by conventional gravimetric method, (b) by modified gravimetric method.**

Revised Figure 1:

[Figure]

**Figure 1. Schematic diagram of the preparation of the acetonitrile gas mixtures with the target amount fractions in specified cylinders (a) by conventional gravimetric method, (b) by conventional gravimetric method for adsorption loss evaluation and (c) by modified gravimetric method. The number in parentheses indicates the number of cylinders prepared at each level.**

**6.** Lines 92–97: Missing information on the manufacturer of DB-1 capillary column. How many replicates of each gas mixture was analysed?

**Authors' response:** In response to the reviewer's comments the subsection 'Analysis of Gas Mixtures' has been revised to improve clarity and added more details, therefore several missing details have been incorporated, and content is as follows:

Revised manuscript (page 5, line 118):

"…DB-1 capillary column (60 m × 0.32 mm, 1 µm film thickness, Agilent, USA) …"

Revised manuscript (page 5, lines 123–125):

"Eight consecutive measurements (i.e., injections) were performed for each sample and peak areas were integrated baseline-to-baseline using the GC software. The averaged peak area for each sample was calculated using at least the last three measurements."

**7.** Equation (1): what was used as reference?

**Authors' response:** The authors revised the subsection "Analysis of Gas Mixtures" to include more details and increase clarity in response to the reviewer's recommendations. As a result, some details that were previously lacking have been included, and content is as follows:

Revised manuscript (page 5, lines 121–123):

"For each preparation level, all gas mixtures were analysed against each other to evaluate the consistency of the gravimetrically prepared gas mixtures, with one of the gas mixtures selected as the working reference."

8. Equations (1) and (2): were the peak areas corrected by the system blank?

**Authors' response:** The blank measurements using high-purity nitrogen showed no detectable peaks; thus, blank correction was not required for the analytical results. The authors now included additional details, as follows:

Original manuscript:

"…Here, $RF_{sample}$ and $RF_{reference}$ represent the response factor of sample and reference, respectively. The uncertainties of response factors were estimated by combining uncertainties from GC analysis and gravimetric preparation…"

Revised manuscript (page 6, lines 145–147):

"…Here, $RF_{sample}$ and $RF_{reference}$ represent the response factor of sample and working reference, respectively, with their standard uncertainties $u(RF_{sample})$ and $u(RF_{reference})$, respectively. Blank measurements using high-purity nitrogen showed no detectable peaks, thus, blank correction was not required for the analytical results…"

9. Lines 117–118: the meaning of the different colours (white, grey, black) should be added to the caption of Figure 2, as well as the meaning of 'RF'. Does the second y-axis on the top panel indicate also the normalized RF?

**Authors' response:** In response to the reviewer's comment, the caption of Figure 2 has been revised. The meaning of 'RF' is included in the caption, and the second y-axis on the top panel also indicates the normalized RF described in the revised caption, as follows:

Original caption:

"Figure 2. Verification results of acetonitrile and hexane gas mixtures less than 10 nmol mol-1. The error bars show expanded uncertainties (k = 2). The Experis (rectangle), Performax (circle), and Untreated (triangle) cylinders."

Revised caption (page 6, lines 160–163):

"Figure 2. Verification results of (a) acetonitrile and (b) hexane gas mixtures at 1 nmol mol$^{-1}$ (white), 3 nmol mol$^{-1}$ (grey), 5 nmol mol$^{-1}$ (dark grey), and 10 nmol mol$^{-1}$ (black) in the Experis (rectangle), Performax (circle), and Untreated (triangle) cylinders. Note that the y-axis and x-axis represents the normalized response factor (RF) and cylinder numbers, respectively, and the error bars show expanded uncertainties ($k = 2$)."

10. A reference or description of the cylinder-to-cylinder division method 29 (lines 120–121) should be added.

**Authors' response:** As per the reviewers' comment, the authors added a reference to the revised manuscript, as follows:

Original manuscript:

> "…we employed the cylinder-to-cylinder division method29 for 3 nmol/mol gas mixtures prepared in all three types of cylinders…"

Revised manuscript (page 7, lines 165–166):

> "…we employed the cylinder-to-cylinder division method (Lee et al., 2017) for 3 nmol mol$^{-1}$ gas mixtures prepared in all three types of cylinders."

**11.** Line 130: the figure numberings should be revised (currently, there are two Figures 1 and none Figure 3). What is the explanation for the large uncertainty of the black triangle point? For comparison purposes, it would be good to plot also the results of the internal standard (hexane). In this figure, results for cylinders at 200 nmol mol$^{-1}$ are shown; it would be good if these cylinders are included in the schematic of Figure 1. Were these cylinders prepared by the conventional or the modified gravimetric method? Why were only the Performax cylinders used for the gas mixtures at 200 nmol mol$^{-1}$? It would be good to add results of the untreated cylinders at 10 µmol mol$^{-1}$.

> **Authors' response:** As per the reviewer's comment, Figure 3 was incorrectly numbered as Figure 1, the authors have revised the figure numbering accordingly.
>
> Regarding the black triangle cylinder, this cylinder was used as a working reference. To evaluate the uncertainty conservatively, the highest standard uncertainty of repeatability and the largest drift from all repeated injections were considered in determining the overall uncertainty. In this case, drift was identified as the primary source of uncertainty due to adsorption losses, which significantly reduced the response factor and, consequently, increasing the relative uncertainty associated with drift.
>
> In response to the reviewer's suggestion, the authors have added cylinders at 200 nmol mol$^{-1}$ to the schematic in Figure 1. Additionally, a description of the development method, referred to as the 'conventional gravimetric method,' has been included in the figure caption. For the 200 nmol mol$^{-1}$ cylinders, only the Performax cylinders showed consistent results with acetonitrile at 3 nmol mol$^{-1}$, despite some loss (page 8, line 179–181). Therefore, the 200 nmol mol$^{-1}$ gas mixtures were developed exclusively using Performax cylinders.
>
> As suggested by the reviewer, the authors added the adsorption loss results of Untreated cylinders at 10 µmol mol$^{-1}$ to the Figure 3 on the page 8, lines 175–176, revised figure as follows:
>
> Original Figure 3:

[Figure]

**Figure 3.** Adsorption loss results of acetonitrile at various amount fractions. The error bars show expanded uncertainties ($k = 2$). The Experis (rectangle), Performax (circle), and Untreated (triangle) cylinders.

Revised Figure 3:

[Figure]

**Figure 3. Adsorption loss results of (a) acetonitrile and (b) hexane at various amount fractions in the Experis (rectangle), Performax (circle) and Untreated (triangle) cylinders. Note that the y-axis and x-axis represents the normalized response factor (RF) and cylinder numbers, respectively, and the error bars show expanded uncertainties ($k = 2$).**

12. Line 139: what is meant by 'little loss'? Can some figure be given (e.g., < 0.1 %)?

**Authors' response:** As per the reviewer's comment, the authors clarified the manuscript on the page 9, lines 192–193. Further, the authors have revised Figure 3 (page 8, lines 175–176) by adding the hexane adsorption loss results for comparison.

Original manuscript:

"The results showed little loss for both acetonitrile and hexane in the Performax cylinders, whereas approximately 1% to 2% loss of acetonitrile was observed…"

Revised manuscript (page 9, lines 192–194):

"The results showed little loss, which is less than its associated relative analytical uncertainties (0.6 % for acetonitrile and 0.2 % for hexane), concerning both acetonitrile and hexane in the Performax cylinders. In contrast, approximately 1 % to 2 % loss of acetonitrile was observed…"

**13.** What are the x–axis labels of Figures 4–6? It would be good if the authors indicate that they refer to the cylinder number. Adding a table with the cylinder number, the type of treatment and the amount fraction of the cylinder might make the text and figures clearer.

**Authors' response:** In response to the reviewer's comment, the authors have included the explanations of the x-axis on Figures 4–6 and the types of cylinder treatments in the figure captions, as follows:

Original caption:

"Figure 4. Verification results of the 100 nmol mol$^{-1}$ acetonitrile and hexane in Performax cylinders. Error bars show expanded uncertainties."

Revised caption (page 10, lines 210–212):

"Figure 4. Verification results of the 100 nmol mol$^{-1}$ (a) acetonitrile and (b) hexane in Performax cylinders. Note that the y-axis and x-axis represents the normalized response factor (RF) and cylinder numbers, respectively, and the error bars show expanded uncertainties ($k = 2$)."

Original caption:

"Figure 5. Long-term stability of the 100 nmol mol-1 acetonitrile in the Performax cylinders. (a) 10 month-stability (peak area ratios of hexane to acetonitrile); (b) 3 year-stability (RFs are normalized to that of an old gas mixture). Error bars show expanded uncertainties."

Revised caption (page 11, lines 227–230):

"Figure 5. Long-term stability of the 100 nmol mol$^{-1}$ acetonitrile in the Performax cylinders (a) 10 month-stability (peak area ratios of hexane to acetonitrile), (b) 3 year-stability (the *RF*s of new gas mixtures are normalized to that of an old gas mixture). Note that the y-axis and x-axis represents the normalized response factor (RF) and cylinder numbers, respectively, and the error bars show expanded uncertainties ($k = 2$)."

Original caption:

"Figure 6. Long-term stability of the 10 μmol mol$^{-1}$ acetonitrile in the Performax cylinders. Error bars show expanded uncertainties."

Revised caption (page 11, lines 240–241):

"Figure 6. Long-term stability of the 10 μmol mol$^{-1}$ acetonitrile in the Performax cylinders. Note that the y-axis and x-axis represents the normalized response factor (RF) and cylinder numbers, respectively, and the error bars show expanded uncertainties ($k = 2$)."

Technical corrections

There is an inconsistent use of nmol mol$^{-1}$ (also μmol mol$^{-1}$) and nmol/mol (also μmol/mol) throughout the paper, the mol mol$^{-1}$ is preferred but usage should be consistent throughout.

**Authors' response:** As per the reviewer's comment, the inconsistencies regarding the notation have been checked and corrected to nmol mol$^{-1}$ (also µmol mol$^{-1}$) throughout the manuscript.

**RC2: Comments**

We appreciate you reviewing our study and providing these insightful comments. Your recommendations have been taken into consideration as we revised our manuscript. Thank you again for your time and expertise.

1.    Page 3, line 81: the text "…and the modified gravimetric method (Brewer, 2011)." Brewer 2011 is not correct here and this should actually be Brewer et al., 2019 (Anal. Chem 91(8), 5310-5315).

>    **Authors' response:**  As per the reviewers' comment, the authors have revised the manuscript to address the issues with the disorganised references in this section, as follows:
>
>    Original manuscript:
>
>    "…and the modified gravimetric method (Brewer, 2011) …"
>
>    Revised manuscript (page 2, lines 59–60):
>
>    "…and the modified gravimetric method (Brewer et al., 2019) …"

2.    Page 3, line 81: the text "…known to be stable in aluminium cylinders (Rhoderick, 2013, Brewer, 2019) …", the reference to Brewer, 2019 is not correct here. An additional reference that should be included here instead is Rhoderick et al., 2019 (doi: 1525/elementa.366) as this paper demonstrates stability for hexane in the nmol mol$^{-1}$ range in the Experis cylinder being used in this study.

>    **Authors' response:**  The authors have revised the manuscript in response to the reviewers' feedback to fix the previously disorganized references, as follows:
>
>    Original manuscript:
>
>    "Hexane, known to be stable in aluminium cylinders (Rhoderick, 2013; Brewer, 2019) …"
>
>    Revised manuscript (page 2, lines 60–61):
>
>    "Hexane known to be stable in aluminium cylinders (Rhoderick et al., 2019) …"

3.    Page 3, line 23: It was not clear to me what the reference was that is referred to here? It looks like from the plots that one mixture has been chosen as the reference and the other results have been normalised to that selected mixture. This was not clear and the choice for the reference was not clear. Some specific text needs to be included to address this.

>    **Authors' response:**  The authors revised the subsection "Analysis of Gas Mixtures" to provide more details and improve clarity in accordance with the reviewer's comments. Consequently, certain details that were previously absent have been included, as shown in the following:
>
>    Revised manuscript (page 5, lines 121–123):

"For each preparation level, all gas mixtures were analysed against each other to evaluate the consistency of the gravimetrically prepared gas mixtures, with one of the gas mixtures selected as the working reference."

Technical corrections

1.      I suggest removing the ACN and hexane labels from the figure replacing them with A and B labels for the two panels and refer to each in the caption with reference to hexane or ACN.

**Authors' response:** To improve clarity and consistency in the figure labelling, the authors have changed the labelling to (a) and (b) in Figures 2 (lines 159–160), 3 (lines 175–176), 4 (lines 209–210), and 5 (lines 227–230) as recommended by the reviewer, the figures as follows:

Revised Figure 2:

[Figure]

**Figure 2. Verification results of (a) acetonitrile and (b) hexane gas mixtures at 1 nmol mol$^{-1}$ (white), 3 nmol mol$^{-1}$ (grey), 5 nmol mol$^{-1}$ (dark grey), and 10 nmol mol$^{-1}$ (black) in the Experis (rectangle), Performax (circle), and Untreated (triangle) cylinders. Note that the y-axis and x-axis represents the normalized response factor (RF) and cylinder numbers, respectively, and the error bars show expanded uncertainties ($k = 2$).**

Revised Figure 3:

[Figure]

**Figure 3.** Adsorption loss results of (a) acetonitrile and (b) hexane at various amount fractions in the Experis (rectangle), Performax (circle) and Untreated (triangle) cylinders. Note that the y-axis and x-axis represents the normalized response factor (RF) and cylinder numbers, respectively, and the error bars show expanded uncertainties ($k = 2$).

Revised Figure 4:

[Figure]

**Figure 4. Verification results of the 100 nmol mol⁻¹ (a) acetonitrile and (b) hexane in Performax cylinders. Note that the y-axis and x-axis represents the normalized response factor (RF) and cylinder numbers, respectively, and the error bars show expanded uncertainties ($k$ = 2).**

Revised Figure 5:

[Figure]

**Figure 5. Long-term stability of the 100 nmol mol⁻¹ acetonitrile in the Performax cylinders (a) 10 month-stability (peak area ratios of hexane to acetonitrile), (b) 3 year-stability (the *RF*s of new gas mixtures are normalized to that of an old gas mixture). Note that the y-axis and x-axis represents the normalized response factor (RF) and cylinder numbers, respectively, and the error bars show expanded uncertainties ($k$ = 2).**

**2.** There is an inconsistent use of nmol mol⁻¹ (also µmol mol⁻¹) and nmol/mol (also µmol/mol) throughout the paper, the mol mol⁻¹ is preferred but usage should be consistent throughout. Currently both formats are used in the paper.

**Authors' response:** As per the reviewer's comment, the inconsistencies regarding the notation have been checked and corrected to mol mol⁻¹ throughout the manuscript.